# Comparative effects of combined aerobic and resistance training versus high-intensity interval training on insulin resistance, glycaemic control, body composition and quality of life in type 2 diabetes: A 12-week randomised controlled trial

**Sampath Kumar Amaravadi**[1,2]⊕*, **Arthur de Sá Ferreira**[1]⊕, **Patrícia dos Santos Vigário**[1]⊕*

**1** Postgraduate Program in Rehabilitation Sciences, Centro Universitário Augusto Motta (UNISUAM), Rio de Janeiro, Brazil, **2** School of Sport, Exercise and Rehabilitation Sciences, College of Life and Environmental Sciences, University of Birmingham, Birmingham, United Kingdom

⊕ These authors contributed equally to this work.
* patriciavigario@souunisuam.com.br (PdSV); s.k.amaravadi@bham.ac.uk (SKA)

## Abstract

### Background

Exercise training is a cornerstone in managing type 2 diabetes mellitus (T2DM), yet direct comparisons between combined aerobic–resistance training (A+R) and high-intensity interval training (HIIT) across clinical and patient-reported outcomes remain limited.

### Objective

To compare the effects of A+R and HIIT on insulin resistance, glycaemic control, body composition, physical function, and quality of life in adults with T2DM, relative to standard care.

### Design and participants

A single-centre, randomised controlled trial involving 90 participants with T2DM (aged 30–65 years), allocated to A+R, HIIT, or control groups.

### Interventions

A+R consisted of moderate-intensity aerobic and resistance exercises, while HIIT comprised structured interval sessions. Both programs were delivered 3–5 times weekly for 12 weeks. The control group received usual care without structured exercise.

### Measurements

Primary outcomes included fasting insulin (FI), Glycosylated Hemoglobin (HbA1c), and insulin resistance (HOMA-IR). Secondary outcomes included fasting glucose

**Data availability statement:** All relevant data underlying the results presented in this study are available in the Open Science Framework (OSF) repository at the following DOI: 10.17605/OSF.IO/4Q9XU.

**Funding:** The author(s) received no specific funding for this work.

**Competing interests:** The authors have declared that no competing interests exist.

(FG), 6-minute walk distance (6MWD), subcutaneous and visceral fat, muscle mass, and WHOQOL-BREF domains.

## Results

Compared with control, the HIIT group showed a greater reduction in fasting glucose (Mean Difference [MD] −29.1 mg/dL; 95% CI −41.2 to −17.0) and the A+R group also improved (MD −20.6 mg/dL; 95% CI −31.0 to −10.2). HbA1c was lower versus control in both HIIT (MD −3.35%; 95% CI −4.11 to −2.58) and A+R (MD −3.33%; 95% CI −4.03 to −2.62). Fasting insulin decreased relative to control in HIIT (MD −7.16 mIU/L; 95% CI −10.04 to −4.28) and A+R (MD −8.87 mIU/L; 95% CI −11.77 to −5.97). HOMA-IR improved versus control in A+R (MD −2.33; 95% CI −3.63 to −1.03) with a non-significant trend in HIIT (MD −1.17; 95% CI −2.47 to 0.13). Functional capacity (6-minute walk distance) increased versus control in HIIT (MD +178.9 m; 95% CI 130.5 to 227.4) and A+R (MD +233.6 m; 95% CI 191.8 to 275.5). Body composition favored both interventions: fat-free mass increased (HIIT MD +7.54 kg; 95% CI 4.71 to 10.36; A+R MD +5.96 kg; 95% CI 3.06 to 8.86) while subcutaneous fat (HIIT MD −7.16%; 95% CI −9.33 to −4.99; A+R MD −8.37%; 95% CI −10.65 to −6.09) and visceral fat (HIIT MD −4.70%; 95% CI −5.93 to −3.47; A+R MD −4.58%; 95% CI −5.86 to −3.31) were reduced. Quality of life improved across domains versus control in both groups (e.g., physical domain: HIIT MD +10.29; 95% CI 4.06 to 16.51; A+R MD +13.77; 95% CI 6.62 to 20.91). All results were derived from covariate-adjusted mixed models with multiple comparison corrections (Benjamini–Hochberg FDR, q = 0.05; Bonferroni-adjusted α = 0.002).

## Limitations

Findings are limited to adherent participants, and generalizability is restricted to those without advanced complications. The 12-week duration precludes assessment of long-term sustainability.

## Conclusion

Both HIIT and A+R significantly improved metabolic, functional, and psychosocial outcomes compared with control. HIIT yielded greater benefits for fasting glucose and muscle mass, while A+R conferred broader improvements in HbA1c, fat reduction, and quality of life. These findings support tailoring exercise prescriptions to therapeutic goals and highlight the complementary roles of HIIT and A+R in routine diabetes care.

## Clinical trial registration

The trial is registered with the Clinical Trial Registry of India (reference no: CTRI/2022/04/041762).

## Introduction

Diabetes mellitus (DM) affects millions of people worldwide, and its incidence continues to increase at an alarming rate. According to global projections, the number of diabetic cases will reach 852.5 million by 2050 [1,2]. Type 2 diabetes mellitus (T2DM) accounts for most of these cases and is associated with significant morbidity, mortality, and economic burden. Lifestyle interventions that include structured exercise training, alongside dietary modifications, have consistently been emphasised in the prevention and management of T2DM [3].

Aerobic exercise training, resistance exercise training, and high-intensity interval training (HIIT) represent three widely studied modalities of physical activity, each offering unique physiological benefits relevant to T2DM management. Aerobic exercises, characterised by sustained moderate-intensity activity such as walking or cycling, are well established for improving cardiovascular fitness, glycaemic control, and overall insulin sensitivity [4]. Resistance training, which involves repeated muscle contractions against external resistance, contributes to increases in muscle mass and strength, thereby enhancing glucose uptake and metabolic health [5]. HIIT, which alternates between short bursts of vigorous activity and recovery periods, has been shown to maximise time efficiency while producing substantial improvements in insulin resistance, glycaemic control, and cardiovascular parameters [6,7]. Evidence from a meta-analysis of 14 studies reported a significant reduction in glycated haemoglobin (HbA1c) levels following structured exercise interventions in individuals with T2DM [8]. Exercise has also been shown to restore insulin sensitivity, improve glycaemic control, and reduce the risk of developing diabetes [9]. Furthermore, when combined with dietary change, physical activity reduces body mass index (BMI), lowers triglycerides concentrations, increases high-density lipoprotein cholesterol levels, and helps regulate blood pressure in individuals at risk of or living with T2DM [10]. Beyond metabolic health, regular physical activity reduces all-cause mortality, lowers cardiovascular and cancer risk, and provides additional benefits by reducing systemic inflammation and oxidative stress [11].

Insulin resistance, defined as the impaired ability of endogenous or exogenous insulin to stimulate glucose uptake, plays a central role in the development of T2DM as part of the broader metabolic syndrome [12]. Contributing factors include visceral adiposity, genetic alterations in insulin signalling pathways, and dietary imbalances. Exercise has been demonstrated to significantly improve insulin sensitivity and glucose tolerance, with even acute bouts inducing favourable metabolic changes [13]. Regular moderate-intensity activity reduces adiposity and enhances cellular insulin responsiveness, partly through the upregulation and translocation of GLUT4 transporters, along with associated skeletal muscle adaptations [14,15]. These changes can sustain insulin-stimulated glucose uptake for several hours post-exercise and improve hepatic glucose and lipid metabolism [16,17]. Such mechanisms provide strong support for structured physical activity as an effective non-pharmacological strategy to improve insulin resistance and reduce cardiometabolic risk in T2DM.

Quality of life (QoL) is also a critical outcome in T2DM management, encompassing physical, psychological, social, and environmental dimensions. Pathophysiological factors such as poor glycaemic control, vascular complications, and associated comorbidities can adversely affect QoL in individuals with T2DM. Lifestyle interventions, particularly structured exercise programmes, have been shown to mitigate these effects [9,18,19]. Exercise may improve functional capacity, mood, and social engagement, thereby contributing to better self-perceived health. Evidence from our previous randomised controlled trial [18] and other reports demonstrates that structured exercise interventions, especially those combining aerobic and resistance modalities, are associated with significant improvements across multiple domains of QoL in individuals with T2DM [9,19]. These findings support the role of exercise as a holistic strategy that addresses not only clinical markers but also patient-centred outcomes [20,21].

Despite the well-documented benefits of exercise for improving insulin sensitivity, glycaemic control, and cardiovascular health, important research gaps remain. Many previous studies have been of short duration, often fewer than 12 weeks, which restricts understanding of the long-term sustainability of exercise-induced improvements. Furthermore, a considerable number of trials have recruited homogeneous cohorts, focusing primarily on middle-aged adults without adequately

representing variations in sex, age, or comorbidities. These limitations reduce the generalisability of findings to the wider and more diverse T2DM population. For example, Al-mhanna et al. highlighted in their narrative review that most trials of exercise interventions in overweight and obese individuals with T2DM were short-term and conducted in relatively uniform populations [22]. Similarly, Tayebi et al., who investigated a 12-week HIIT intervention in obese men, emphasised the need for studies that include women and older adults as well as longer follow-up durations [23]. Addressing these short-comings is essential to guide clinical practice and inform public health recommendations.

To address these gaps, the present trial was designed to evaluate the comparative effects of two distinct but widely recommended training modalities, combined aerobic and resistance training (A+R) and HIIT, in adults with T2DM. By examining both metabolic outcomes such as fasting glucose, insulin resistance, and HbA1c, and patient-centred outcomes including body composition, functional capacity, and QoL, this study provides a comprehensive evaluation of exercise efficacy. Importantly, the inclusion of a diverse cohort and the use of a 12-week intervention period strengthen the external validity of the findings. In addition, structured exercise programmes such as A+R and HIIT are relatively low-cost and scalable lifestyle interventions that may enhance accessibility in diverse healthcare contexts, as highlighted by recent reports on circuit and aquatic-based programmes [24, 25]. Integrating both clinical and psychosocial outcomes positions this trial to contribute novel evidence that extends beyond surrogate markers and informs personalised and sustainable exercise prescriptions in diabetes care.

Therefore, the objective of this randomised controlled trial was to compare the effects of a 12-week programme of A+R and HIIT, relative to standard care, on insulin resistance, glycaemic control, body composition, functional capacity, and QoL in adults with T2DM. We hypothesised that both exercise interventions would significantly improve metabolic and psychosocial outcomes compared with control, and that HIIT would confer greater improvements in fasting glucose and muscle mass, whereas A+R would demonstrate broader benefits in HbA1c, adiposity reduction, and QoL domains.

## Materials and methods

### Design

This was a 12-week, single-center randomized controlled trial conducted between 2 June 2021 and 30 November 2022. Ethical approval was obtained from the Institutional Ethics Committee (IRB/COHS/FAC/20/June2021), and written informed consent was provided by all participants prior to enrolment. The trial was prospectively registered with the Clinical Trials Registry of India (CTRI/2022/04/041762). The study was conducted and reported in accordance with the Consolidated Standards of Reporting Trials (CONSORT) guidelines for randomised controlled trials, including the updated 2025 recommendations [26].

### Setting

The exercise intervention was performed at a tertiary care hospital in Ajman, United Arab Emirates (UAE).

### Participants

Eligible participants included individuals who had been diagnosed with T2DM and were currently receiving treatment with oral hypoglycaemic agents with or without insulin therapy. The age range for inclusion was 30–65 years, and both males and females were eligible to participate. Participants meeting any of the following criteria were excluded from the study: confirmed respiratory disease, coronary artery disease, neurological disorders, musculoskeletal problems that could hinder exercise training, uncontrolled hypertension (systolic blood pressure > 180 mmHg or diastolic blood pressure > 120 mmHg), pregnancy, thyroid disorders, or lack of willingness to participate in the study. The purpose of the study and the benefits of participation were explained to the prospective participants. The participants were asked to read an information sheet and sign a written informed consent form.

A total of 200 participants underwent screening, and 90 eligible participants were ultimately recruited based on meeting the inclusion and exclusion criteria. Initial screening involved the use of the Physical Activity Readiness Questionnaire (PAR-Q) [27] and American Heart Association (AHA)/American College of Sports Medicine (ACSM) risk stratification questionnaire [28]. Eligible participants were categorized into a combined aerobic and resistance exercise training (A+R) group (A) (n = 30), a HIIT programme group (B) (n = 30), and a control group (C) (n = 30) (Fig 1) [26].

## Randomization

Following evaluation by a physiotherapist, participants were randomly allocated to one of three groups: combined aerobic and resistance training (A+R), high-intensity interval training (HIIT), or control. The A+R group undertook an individually tailored, structured, and supervised programme combining aerobic and resistance exercises. The HIIT group performed structured high-intensity interval sessions, while the control group continued with standard care without structured exercise. Block randomization was employed using Sequentially Numbered Opaque Sealed Envelopes (SNOSE), with the principal investigator generating the allocation sequence and assigning participants according to the envelope draw. Assessments and exercise prescriptions were provided free of charge to all participants. The study used nine blocks, with 10 participants per block.

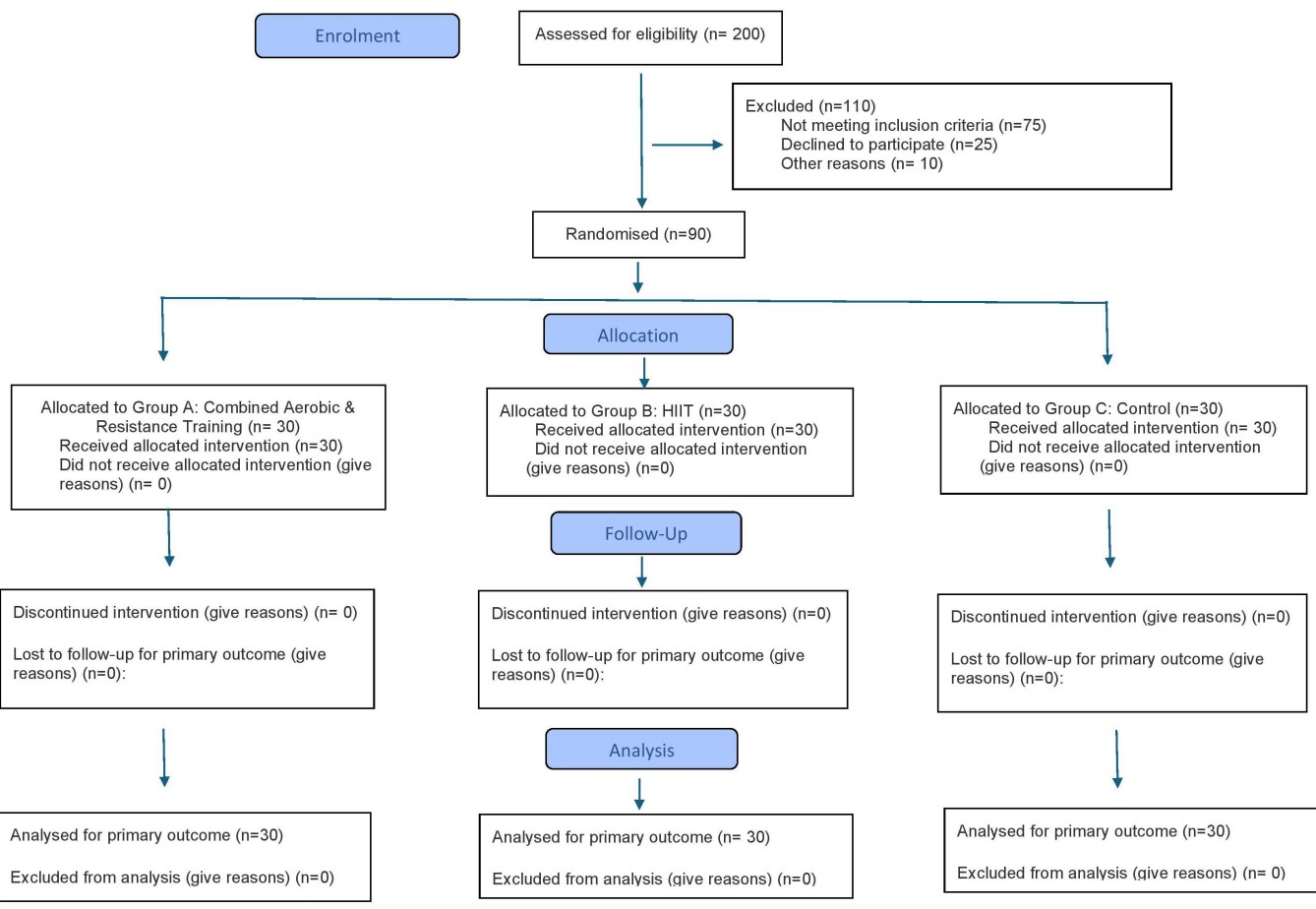

**Fig 1. This is the Fig 1 Consort flowchart of participants.**

## Intervention

Group A participated in a 12-week intervention that included an individualised combined aerobic and resistance exercise training program along with standard hospital care. Group B underwent a HIIT programme program adapted from the model described by Weston et al. [29], which typically involved repeated short intervals of high-intensity exercise interspersed with recovery periods. To suit the safety and physical capacity of participants with T2DM, the protocol was modified as follows: (1) the intensity was adjusted to 85–90% of peak heart rate rather than maximal effort; (2) interval durations were standardized to 1-minute high-intensity bouts followed by 1-minute active recovery, repeated for 10 cycles per session; and (3) a longer warm-up and cool-down phase (5 minutes each at 50–60% HRmax) was added. These modifications aimed to optimize safety, adherence, and tolerability while maintaining the physiological demands consistent with HIIT principles. Both programs incorporated activities such as walking and exercises for the upper and lower limbs, tailored to each participant based on their baseline six-minute walk distance (6MWD) and rating of perceived exertion (RPE). Exercises were demonstrated to the patients and their caregivers during the initial visit, with a patient education manual provided for guidance.

For the first two weeks, the participants engaged in supervised sessions within the hospital. Following the initial supervised sessions, participants transitioned to a structured home-based HIIT programme with predefined progressions implemented at weeks 2 and 6. Each session began with a 10-minute warm-up of light aerobic exercises, followed by interval walking alternating between 1-minute high-intensity bouts (at an RPE of 7–8 out of 10) and 1-minute active recovery periods (at an RPE of 3–4). The total duration of the interval phase progressed from 20 minutes in the first two weeks to 30 minutes by week 6, and up to 45 minutes by the end of the intervention. Sessions concluded with a 10-minute cool-down. All HIIT sessions were monitored using Polar heart rate monitors, and intensity was prescribed relative to each participant's maximum heart rate (HRmax). Progression was guided by weekly physiotherapist supervision and confirmed using the Borg RPE scale to ensure safe and effective workload adjustments. Participants were advised to exercise 3–5 times weekly and to recognise signs of hypoglycaemia and other exercise termination criteria, such as chest pain or shortness of breath. All supervised sessions were conducted in the late afternoon (between 3:00 and 7:00 pm), and participants were encouraged to maintain this schedule for their home-based sessions to ensure consistency across groups. This timing was selected to accommodate work schedules and because emerging evidence suggests that afternoon or evening exercise may enhance glycaemic control more effectively than morning exercise in individuals with T2DM [30–32].

The exercise programme targeted major muscle groups and was initially prescribed as 1–2 sets of 5 repetitions, with both the number of sets and repetitions progressively increased every two weeks according to participant tolerance. By the end of the 12-week programme, individual session durations ranged from 30 to 40 minutes. Adherence was recorded as the proportion of supervised sessions attended, with participants also maintaining logbooks and receiving weekly follow-up calls to encourage engagement (≥70% attendance was required). Compliance was defined as the proportion of attended sessions in which the prescribed training intensity (target heart rate and Borg RPE scale) was achieved. Attendance was documented by supervising physiotherapists, and compliance was verified through heart rate monitor recordings

The control group received standard hospital care without a structured exercise program. All the participants were evaluated at baseline and after 12 weeks. For illiterate patients, caregivers assisted them in understanding and adhering to the program. The exercise protocols were designed based on Maiya et al. (2006) [33] and Amaravadi [18] aligned with the American College of Sports Medicine (ACSM) guidelines [34]. Intervention details are summarized according to the FITT-VP framework [35] in Appendix 1 to improve clarity and reproducibility.

## Outcomes and measurements

The primary outcome measures in this study included fasting insulin levels (FI) [mIU/L], Homeostatic Model Assessment of Insulin Resistance (HOMA-IR), and quality of life assessed using the World Health Organization Quality of Life-BREF (WHOQOL-BREF) questionnaire (Cronbach's α = 0.73–0.81) [36]. The secondary outcome measures included functional

aerobic capacity, assessed by using the six-minute walk test (6MWT) with the outcome expressed as distance walked in meters; body composition parameters, including subcutaneous fat, visceral fat, and muscle mass, were assessed using a bioelectrical impedance analyzer (Tanita MC-780MA, Tanita Corporation, Tokyo, Japan), fasting glucose levels (FG) [mg/dL], postprandial blood sugar levels (PPBS) [mg/dL], HbA1c levels [%], assessment of physical activity using the Global Physical Activity Questionnaire (GPAQ) (Cronbach's α = 0.83–0.96) [37]. Waist and hip circumferences were measured using a flexible, non-stretchable tape measure following the standardised protocols of the International Society for the Advancement of Kinanthropometry (ISAK). Measurements were taken twice, and the mean value was used for analysis [38]. The three groups were assessed using these outcome measures at baseline and after the 12-week intervention period. All outcome assessments were conducted in the morning following an overnight fast of at least 10 hours. Participants were instructed to refrain from vigorous physical activity and caffeine consumption for at least 24 hours before testing to reduce potential confounding effects. All outcome assessors were blinded to group allocation throughout the study. The research staff responsible for conducting clinical, functional, and questionnaire-based assessments were not involved in the intervention delivery and had no access to participant randomization information, thereby minimizing the risk of detection bias.

Adverse events were monitored throughout the intervention by the principal investigator, who supervised all exercise sessions. Participants were instructed to report any injuries, hypoglycaemic episodes, or other untoward effects. All events were documented using structured reporting forms and reviewed weekly by the principal investigator. Adherence was tracked using participant logbooks and weekly follow-up calls.

## Statistical analysis

Baseline characteristics were summarised as mean ± standard deviation (SD) for continuous variables and n (%) for categorical variables. Between-group comparisons were examined using one-way analysis of variance (ANOVA) for continuous variables and Chi-square ($\chi^2$) tests for categorical variables. Given the relatively small sample size and the large number of baseline variables, *P*-values were adjusted using both the Benjamini–Hochberg False Discovery Rate (FDR, q = 0.05) and Bonferroni correction (m = 23 tests; adjusted α = 0.002). These adjusted *P*-values are presented in Table 1 to address reviewer concerns regarding multiple comparisons. Consistent with CONSORT 2025 guidance [26], these results are descriptive and were not used to determine baseline equivalence; instead, baseline imbalances were subsequently included as covariates in the linear mixed-effects models.

Statistical analyses were performed using the RCTapp Shiny interface [39]. The primary analysis was conducted using a linear mixed-effects model with participants specified as random effects and fixed factors for group (HIIT, A + R, Control), time (Pre, Post), and the group × time interaction. To account for observed baseline imbalances, models were covariate-adjusted for age, duration of type 2 diabetes mellitus (T2DM), baseline fasting glucose (FG), and baseline WHOQOL-BREF physical domain scores. This adjustment ensured that estimated treatment effects reflected true intervention-related changes rather than pre-existing group differences. Two-sided *P*-values < 0.050 were considered statistically significant. Between-group mean differences and corresponding 95% confidence intervals (CIs) from the covariate-adjusted model were reported for interpretability. Given the number of primary and secondary outcomes, the potential for Type I error inflation was acknowledged. Although *P*-values for inferential outcomes (Table 3) were not formally adjusted, interpretation emphasised effect sizes and confidence intervals over nominal significance. This approach aligns with PLOS ONE and CONSORT 2025 guidance, which prioritise estimation-based reporting over strict multiplicity correction in exploratory analyses.

## Sample size calculation

Sample size estimation was conducted using G*Power version 3.1.9.7, based on a repeated-measures ANOVA (within-between interaction) to detect changes in clinical outcomes across three groups: high-intensity interval training (HIIT), combined aerobic and resistance training (A + R), and a control group receiving standard care. The calculation assumed a moderate effect size (f = 0.30), with an alpha level of 0.05, power of 0.80, two measurement time points (baseline and 12

**Table 1. Baseline Demographic and Clinical Characteristics of Participants (Adjusted for Multiple Comparisons).**

| Variable | A+R (n=30) Mean±SD | HIIT (n=30) Mean±SD | Control (n=30) Mean±SD | P (unadjusted) | P_adj (FDR) | P_adj (Bonferroni) |
|---|---|---|---|---|---|---|
| Gender, n (%) | 16 (53.3)/ 14 (46.7) | 18 (60.0)/ 12 (40.0) | 14 (46.7)/ 16 (53.3) | P=0.73 | 0.730 | 1.000 |
| Age (years) | 55.1±6.2 | 45.9±10.3 | 50.4±8.5 | P=0.002 | **0.005** | 0.046 |
| T2DM duration (years) | 16.7±9.5 | 7.1±4.1 | 9.6±6.0 | P<0.001 | **0.001** | 0.011 |
| Waist circumference (cm) | 79.8±14.9 | 89.0±12.1 | 85.4±7.7 | P=0.037 | 0.053 | 0.851 |
| Hip circumference (cm) | 100.2±11.8 | 92.7±11.5 | 96.2±9.0 | P=0.047 | 0.064 | 1.000 |
| BMI (kg/m²) | 25.1±3.7 | 23.1±1.8 | 23.5±3.9 | P=0.12 | 0.138 | 1.000 |
| OHA, n (%) | 21 (70.0) | 30 (100.0) | 22 (73.3) | P=0.004 | **0.008** | 0.092 |
| Insulin therapy, n (%) | 4 (13.3) | 3 (10.0) | 10 (33.3) | P=0.08 | 0.097 | 1.000 |
| FG (mg/dL) | 152.3±30.8 | 180.9±34.9 | 126.9±28.6 | P<0.001 | **0.001** | 0.011 |
| PPBS (mg/dL) | 221.7±30.5 | 199.3±49.6 | 135.7±35.7 | P<0.001 | **0.001** | 0.011 |
| HbA1c (%) | 8.5±1.2 | 9.4±1.6 | 7.8±1.5 | P=0.001 | **0.003** | 0.023 |
| FI (µU/mL) | 16.6±5.6 | 12.7±3.5 | 14.3±6.8 | P=0.07 | 0.089 | 1.000 |
| HOMA-IR | 6.1±3.4 | 6.2±5.3 | 4.3±1.9 | P=0.19 | 0.208 | 1.000 |
| Subcutaneous fat (%) | 29.2±5.8 | 26.8±5.1 | 24.4±5.4 | P=0.012 | **0.021** | 0.276 |
| Visceral fat | 13.7±3.0 | 11.6±3.2 | 11.7±3.2 | P=0.03 | **0.046** | 0.690 |
| Muscle mass (kg) | 19.6±4.4 | 20.5±4.4 | 24.6±4.6 | P<0.001 | **0.001** | 0.011 |
| 6MWT distance (m) | 444.3±69.5 | 496.7±51.4 | 368.3±76.5 | P<0.001 | **0.001** | 0.011 |
| WHOQOL-BREF Physical | 62.1±9.3 | 54.2±12.6 | 45.2±6.7 | P<0.001 | **0.001** | 0.011 |
| WHOQOL-BREF Psychological | 60.8±12.1 | 48.4±11.3 | 50.2±11.4 | P<0.001 | **0.001** | 0.011 |
| WHOQOL-BREF Social | 63.0±12.0 | 54.7±19.1 | 52.1±16.9 | P=0.03 | **0.046** | 0.690 |
| WHOQOL-BREF Environmental | 78.1±15.2 | 54.3±11.9 | 51.9±13.9 | P<0.001 | **0.001** | 0.011 |
| Physical activity (MET-min/wk) | 1204±1308 | 962±683 | 745±1019 | P=0.25 | 0.261 | 1.000 |
| Sedentary time (min/wk) | 3607±1297 | 3853±1135 | 2654±880 | P=0.006 | **0.011** | 0.138 |

Values are presented as mean±SD for continuous variables and n (%) for categorical variables. Between-group differences were examined using one-way ANOVA or Chi-square tests. *P*-values were adjusted for multiple comparisons using both the Benjamini–Hochberg False Discovery Rate (FDR; q=0.05) and Bonferroni correction (m=23 tests; adjusted α=0.002). Only variables meeting FDR significance are bolded. Baseline imbalances were subsequently adjusted for in the mixed-effects model to ensure robust estimation of intervention effects.

Given the large number of comparisons relative to the sample size (n=30 per group), these baselines differences should be interpreted descriptively. Baseline imbalances were subsequently adjusted for in the linear mixed-effects models (see Statistical Analysis section).

weeks post-intervention), and an assumed correlation of 0.5 between repeated measures. These parameters indicated that a minimum of 66 participants (22 per group) would be required to detect significant group-by-time interaction. To account for an anticipated 20% dropout rate, the final target sample size was adjusted to 84 participants (28 per group). The assumed moderate effect size (f=0.30) was based on prior trials of structured exercise interventions in T2DM, which reported mean effects ranging from f=0.25–0.35 for HOMA-IR and HbA1c outcomes [18,40,41]. This value was therefore considered clinically meaningful and realistic for the anticipated metabolic improvements.

## Results

A total of 90 participants were randomised into three groups (A+R: n=30; HIIT: n=30; Control: n=30), and all participants completed the 12-week intervention and follow-up assessments. As there was no attrition, a per-protocol analysis was conducted. Details of recruitment, allocation, follow-up, and analysis are presented in the CONSORT flow diagram (Fig 1). Adherence to the exercise interventions was high, with participants in the A+R group attending 92% of sessions and

those in the HIIT group attending 94%. Compliance with prescribed intensity was 88% in A+R and 91% in HIIT, confirmed by HR and RPE monitoring.

## Characteristics of participants

Table 1 summarizes the baseline demographic, clinical, glycaemic, and functional characteristics of participants across the three groups: Combined Aerobic+Resistance (A+R), High-Interval Training (HIIT), and Control (n=30 each). Values are presented as mean±standard deviation (SD) for continuous variables and n (%) for categorical variables. Between-group comparisons were performed using one-way analysis of variance (ANOVA) for continuous variables and Chi-square ($\chi^2$) tests for categorical variables. To control for the increased risk of Type I error arising from the large number of comparisons relative to the modest sample size, P-values were adjusted using both the Benjamini–Hochberg False Discovery Rate (FDR, q=0.05) and the Bonferroni correction (α=0.002, m=23 tests). After adjustment, only a subset of variables remained statistically significant under the FDR threshold, specifically the duration of type 2 diabetes mellitus (T2DM), fasting glucose (FG), post-prandial blood sugar (PPBS), muscle mass (MM), six-minute walk distance (6MWD), and WHOQOL-BREF physical, psychological, and environmental domains. These findings reflect true baseline imbalances, which were subsequently included as covariates in the mixed-effects model to ensure accurate estimation of intervention effects. Table 2 provides a summary of comorbidities and diabetes-related health behaviors.

**HIIT:** high-intensity interval training; **OHA:** oral hypoglycaemic agent. **BMI:** body mass index; **T2DM:** type 2 diabetes mellitus**; HbA1c:** glycated haemoglobin; **FG**: fasting glucose; **PPBS:** postprandial blood sugar; **HOMA-IR:** homeostatic model assessment of insulin resistance; **6MWT:** six-minute walk test; **GPAQ:** Global Physical Activity Questionnaire.

**Fasting glucose (FG):** Compared with Control, HIIT reduced fasting glucose (MD −29.09 mg/dL, 95% CI −41.16 to −17.02) and A+R also improved glycaemic control (MD −20.59 mg/dL, 95% CI −30.96 to −10.21). The A+R vs. HIIT contrast was small and imprecise (MD 8.51 mg/dL, 95% CI −2.01 to 19.02).

**HbA1c**: Both interventions lowered HbA1c relative to Control: HIIT (MD −3.35%, 95% CI −4.11 to −2.58) and A+R (MD −3.33%, 95% CI −4.03 to −2.62). A+R vs. HIIT showed no meaningful difference (MD 0.02%, 95% CI −0.69 to 0.73).

**Fasting insulin (FI) and insulin resistance (HOMA-IR):** Fasting insulin decreased vs. Control in HIIT (MD −7.16 mIU/L, 95% CI −10.04 to −4.28) and A+R (MD −8.87 mIU/L, 95% CI −11.77 to −5.97). HOMA-IR improved in A+R relative to Control (MD −2.33, 95% CI −3.63 to −1.03), with a non-significant trend favoring HIIT (MD −1.17, 95% CI −2.47 to 0.13).

**Post-prandial blood sugar (PPBS):** Between-group estimates did not show a beneficial effect for HIIT vs. Control (MD 5.35 mg/dL, 95% CI −19.76 to 30.47) and indicated worse PPBS for A+R relative to Control (MD 36.11 mg/dL, 95%

**Table 2. Baseline Comorbidities and Health Behaviors of Participants.**

| Particulars | A+R group (n=30) | HIIT group (n=30) | Control group (n=30) |
|---|---|---|---|
| Overweight | 8 (26.7) | 0 (0.0) | 9 (30.0) |
| Hypertension | 19 (63.3) | 21 (70.0) | 14 (46.7) |
| Eye complications | 15 (50.0) | 0 (0.0) | 0 (0.0) |
| Smoking | 17 (56.7) | 4 (13.3) | 10 (33.3) |

Values are n (%)**; A+R:** Combined Aerobic–Resistance training; **HIIT:** High-Intensity Interval Training.

**Insulin resistance, glycaemic control, functional capacity, and quality of life.**

A+R vs. HIIT Group vs. Control Group using linear mixed models adjusted for baseline covariates, significant group×time interactions were observed across metabolic, functional, and psychosocial outcomes. Both A+R and HIIT outperformed Control on the primary glycaemic outcomes, with generally comparable effects between the two active interventions (Table 3).

**Table 3. Changes in metabolic, body composition, and quality-of-life outcomes following 12 weeks of exercise intervention.**

| Outcome (units) | HIIT Within-group Δ (Post – Pre) | A+R Within-group Δ (Post – Pre) | Control Within-group Δ (Post – Pre) | HIIT vs Control MD (95% CI) | A+R vs Control MD (95% CI) | A+R vs HIIT MD (95% CI) |
|---|---|---|---|---|---|---|
| FI | −5.6±3.9* | −5.8±4.1* | −0.9±3.5 | −7.16 (−10.04 to −4.28) | −8.87 (−11.77 to −5.97) | −1.71 (−4.69 to 1.28) |
| FG | −29±22* | −21±20* | −1±18 | −29.09 (−41.16 to −17.02) | −20.59 (−30.96 to −10.21) | 8.51 (−2.01 to 19.02) |
| HbA1c | −0.7±0.5* | −0.8±0.5* | −0.1±0.4 | −3.35 (−4.11 to −2.58) | −3.33 (−4.03 to −2.62) | 0.02 (−0.69 to 0.73) |
| HOMA-IR | −1.3±1.0* | −1.4±1.0* | −0.2±0.9 | −1.17 (−2.47 to 0.13) | −2.33 (−3.63 to −1.03) | −1.16 (−2.44 to 0.11) |
| PPBS | −27±42* | −36±43* | −2±39 | 5.35 (−19.76 to 30.47) | 36.11 (7.97 to 64.26) | 30.76 (9.34 to 52.19) |
| 6MWD | +177±68* | +233±64* | +7±53 | 178.91 (130.47 to 227.35) | 233.61 (191.75 to 275.47) | 54.71 (14.92 to 94.50) |
| Fat-Free Mass (kg) | +7.6±5.2* | +6.0±5.0* | +0.4±4.9 | 7.54 (4.71 to 10.36) | 5.96 (3.06 to 8.86) | −1.58 (−4.22 to 1.07) |
| Subcutaneous Fat (%) | −7.2±4.0* | −8.3±4.1* | −0.5±3.8 | −7.16 (−9.33 to −4.99) | −8.37 (−10.65 to −6.09) | −1.21 (−3.38 to 0.96) |
| Visceral Fat (%) | −4.7±2.8* | −4.6±3.0* | −0.2±2.7 | −4.70 (−5.93 to −3.47) | −4.58 (−5.86 to −3.31) | 0.11 (−1.17 to 1.39) |
| QoL – Physical (WHOQOL-BREF) | +10.3±6.2* | +13.7±6.1* | +0.5±5.8 | 10.29 (4.06 to 16.51) | 13.77 (6.62 to 20.91) | 3.48 (−2.65 to 9.61) |
| QoL – Psychological | +17.3±7.4* | +27.8±7.2* | +0.7±6.8 | 17.26 (10.55 to 23.97) | 27.80 (20.64 to 34.96) | 10.54 (3.21 to 17.87) |
| QoL – Social | +23.9±8.3* | +32.2±8.1* | +0.5±7.2 | 23.93 (16.53 to 31.34) | 32.20 (24.53 to 39.87) | 8.26 (0.71 to 15.82) |
| QoL – Environmental | +39.6±9.6* | +43.3±9.8* | +0.7±8.7 | 39.60 (33.51 to 45.69) | 43.37 (35.62 to 51.13) | 3.78 (−3.71 to 11.27) |
| Sedentary Time (min/week) | −653±398* | −429±374* | −5±326 | −653.42 (−1087.96 to −218.87) | −428.81 (−849.88 to −7.73) | 224.61 (−173.99 to 623.21) |
| Physical Activity (MET-min/week) | +845±372* | +1138±358* | +3±340 | 845.29 (292.06 to 1398.51) | 1137.51 (577.09 to 1697.93) | 292.23 (−261.51 to 845.96) |

Values are presented as estimated marginal means±standard error (SE) derived from the covariate-adjusted linear mixed-effects model. Between-group differences reflect post-intervention effects adjusted for baseline values, age, duration of T2DM, FG, and WHOQOL-BREF physical domain scores.

P-values are presented for primary inferential comparisons and were not further adjusted for multiple outcomes, as the mixed-model analysis accounts for repeated measures and correlated outcomes. However, the potential for Type I error inflation due to multiple comparisons was considered during interpretation, and results are presented alongside corresponding effect sizes and 95% confidence intervals for transparency. *.

**FI**: Fasting Insulin (mIU/L); **FG**: Fasting glucose (mg/dL); **HbA1c**: Glycated Haemoglobin (%); **HOMA-IR**: Homeostatic Model Assessment of Insulin Resistance; **PPBS**: Post-Prandial Blood Sugar (mg/dL); **6MWD**: Six-Minute Walk Distance (metres); **GPAQ**: Global Physical Activity Questionnaire (MET-min/week); **SED**: Sedentary Time (minutes/week); **HIIT**: High-Intensity Interval Training; **A+R**: Combined aerobic and resistance exercise training.

CI 7.97 to 64.26); A+R vs. HIIT favoured HIIT (MD −30.76 mg/dL, 95% CI −52.19 to −9.34). These findings should be interpreted in the context of broader glycaemic improvements (fasting glucose, HbA1c, fasting insulin) that consistently favored both active interventions.

**Functional capacity (6-minute walk distance):** Compared with Control, HIIT improved to 6MWD (MD+178.91 m, 95% CI 130.47 to 227.35) and A+R yielded a larger gain (MD+233.61 m, 95% CI 191.75 to 275.47). A+R exceeded HIIT (MD+54.71 m, 95% CI 14.92 to 94.50).

**Body composition:** We observed favourable between-group effects for both interventions versus Control. Fat-free mass increased with HIIT (MD+7.54 kg, 95% CI 4.71 to 10.36) and A+R (MD+5.96 kg, 95% CI 3.06 to 8.86); subcutaneous fat (HIIT MD −7.16%, 95% CI −9.33 to −4.99; A+R MD −8.37%, 95% CI −10.65 to −6.09) and visceral fat (HIIT MD −4.70%, 95% CI −5.93 to −3.47; A+R MD −4.58%, 95% CI −5.86 to −3.31) decreased. The A+R–HIIT contrasts were small with CIs crossing zero, indicating no clear difference between the active interventions.

**Physical activity and sedentary time:** Compared with Control, total physical activity (GPAQ MET-min/week) increased for HIIT (MD+845.29, 95% CI 292.06 to 1398.51) and A+R (MD+1137.51, 95% CI 577.09 to 1697.93). Sedentary time decreased for HIIT (MD −653.42 min/week, 95% CI −1087.96 to −218.87) and A+R (MD −428.81 min/week, 95% CI −849.88 to −7.73). The A+R vs HIIT contrasts for these behavioral outcomes were imprecise.

**Quality of life (WHOQOL-BREF):** All four domains improved versus Control in both interventions. Representative effects include the physical domain (HIIT MD + 10.29, 95% CI 4.06 to 16.51; A + R MD + 13.77, 95% CI 6.62 to 20.91), psychological (HIIT MD + 17.26, 95% CI 10.55 to 23.97; A + R MD + 27.80, 95% CI 20.64 to 34.96), social (HIIT MD + 23.93, 95% CI 16.53 to 31.34; A + R MD + 32.20, 95% CI 24.53 to 39.87), and environmental (HIIT MD + 39.60, 95% CI 33.51 to 45.69; A + R MD + 43.37, 95% CI 35.62 to 51.13).

## Adverse events

No adverse events were reported by participants in either intervention group. Adherence to the exercise programme was defined as completion of >70% of prescribed sessions, and most participants met this criterion.

## Discussion

This study comprehensively evaluated the comparative effects of combined aerobic and resistance exercise training (A + R) and high-intensity interval training (HIIT) on a broad array of clinical, functional, and psychosocial outcomes in adults with T2DM. Both interventions produced significant within-group improvements from baseline in glycaemic control, insulin sensitivity, body composition, physical functioning, and quality of life. Between-group analyses further demonstrated that participants in the A + R and HIIT groups achieved greater improvements than the control group across these domains. Importantly, the magnitude and nature of change differed between interventions: HIIT was more effective in reducing fasting blood sugar and improving environmental quality of life, whereas A + R yielded greater benefits in HbA1c, insulin resistance, and social and physical quality-of-life domains. These findings highlight the value of structured exercise in diabetes care while supporting an individualized approach to exercise prescription. They are consistent with emerging evidence advocating tailored lifestyle interventions in the management of chronic metabolic conditions.

Several potential confounders warrant consideration when interpreting these findings. Dietary intake was not systematically monitored, which may have influenced glycaemic and lipid outcomes, consistent with the interaction between diet quality and exercise responses reported in contemporary syntheses [22]. Unrecorded changes in antihyperglycemic medication, habitual sleep duration, and daily activity outside supervised sessions could also have contributed to variability. Most sessions occurred in the late afternoon, and time of day may modulate glycaemic responses and lipid handling according to recent evidence [30,32]. Adherence varied between participants, and although our per-protocol and intention-to-treat analyses converged, greater session attendance was associated with larger within-group improvements. Finally, body composition was assessed by bioimpedance rather than DXA, which may underestimate small changes in regional adiposity and subcutaneous fat.

### Blood glucose regulation and insulin sensitivity

Both exercise groups showed significant within-group improvements from baseline in key glycaemic outcomes, including fasting blood sugar, postprandial blood sugar, fasting insulin, and HOMA-IR. Between-group analyses indicated that these improvements were greater in the A + R and HIIT groups than in the control group, confirming the efficacy of structured exercise for metabolic control in T2DM. Direct comparisons between A + R and HIIT did not identify statistically significant differences on these primary glycaemic outcomes, indicating no clear superiority of one modality over the other. Magnitudes and precision of effects are reported as Cohen's d with 95% confidence intervals in the Results and in Table 3.

For fasting and postprandial blood sugar, HIIT exhibited numerically larger within-group reductions, consistent with mechanisms such as enhanced mitochondrial biogenesis, increased GLUT4 translocation, and improved skeletal muscle glucose uptake during repeated high-intensity efforts [42,43]. These observations align with prior reports that interval-based training can elicit rapid glycaemic benefits in time-efficient formats. However, the differences relative to A + R did not reach statistical significance, underscoring that both interventions are comparably effective for glucose regulation.

Both interventions also reduced fasting insulin and improved HOMA-IR compared with baseline and control, supporting enhanced peripheral insulin sensitivity with structured training. Although numerically greater declines in fasting insulin were observed with HIIT, between-group differences were not statistically significant, indicating that either modality can be used to improve insulin dynamics depending on patient preference and feasibility.

HbA1c also decreased significantly in both intervention arms, confirming improvements in long-term glycaemic regulation. A+R produced slightly greater absolute reductions, a profile consistent with the prolonged cardiovascular stimulus and additive resistance component characteristic of combined training. These findings echo evidence that both interval and continuous training lower HbA1c through distinct physiological pathways [44]. The between-group difference, however, was not statistically significant, supporting therapeutic equivalence of A+R and HIIT for this outcome.

Our pattern of effects aligns with recent evidence. The apparent advantage of HIIT for fasting glucose reduction is concordant with meta-analytic findings that higher-intensity interval formats enhance glucose regulation and insulin resistance [44], while the broader benefits observed with combined aerobic and resistance training on HbA1c, adiposity indices, and quality of life echo reports that multi-component programmes confer wider cardiometabolic and psychosocial gains [21,41]. Narrative and trial data published in 2024–2025 extend this view: Al-mhanna et al. highlighted that exercise effects on lipid metabolism in overweight or obese adults with T2DM are sensitive to programme design and participant heterogeneity [22], and Tayebi et al. demonstrated clinically relevant improvements after a 12-week HIIT protocol in obese men, while calling for inclusion of women and older adults to strengthen generalisability [23]. These converging findings support the external validity of our results in diverse service contexts. To aid clinical interpretation, we report effect sizes with 95% confidence intervals for all primary outcomes in the Results and in Table 3, which clarifies the magnitude and precision of treatment effects.

## Quality of life and functional outcomes

Improvements in functional capacity were reflected in significant increases in six-minute walk distance (6MWD) in both exercise groups, underscoring the role of structured activity in enhancing cardiorespiratory fitness and endurance in T2DM patients. A+R elicited greater gains, due to its continuous aerobic stimulus combined with resistance training, which improves muscular efficiency and exercise tolerance. The effect of functional enhancement is especially critical in this population given the link between physical inactivity, mobility limitations, and disease progression.

In terms of quality of life, both interventions produced significant gains across all four WHOQOL-BREF domains. A+R demonstrated superior improvements in physical, psychological, and social domains, suggesting it is particularly effective in promoting holistic well-being. HIIT, on the other hand, showed the greatest benefit in the environmental domain, potentially reflecting improved perceived autonomy, accessibility, and adaptability to lifestyle contexts. These domain-specific gains are congruent with the findings of Collins et al. (2021), who reported enhanced psychosocial outcomes following structured aerobic routines in individuals with chronic illnesses [9]. The social benefits observed with A+R may also stem from its moderate intensity, which may facilitate greater adherence and enjoyment, thereby positively reinforcing self-efficacy and patient motivation.

Our results mirror those of Gajanand et al. (2023), who demonstrated that low-volume combined aerobic and resistance HIIT improved not only physiological markers but also health-related QoL metrics among people living with T2DM [19]. This reinforces the potential for blended training protocols to deliver comprehensive health improvements across both physiological and psychosocial domains.

## Body composition adaptations

Both exercise interventions improved body composition, although the nature of changes differed. A+R produced greater reductions in subcutaneous and visceral fat, which is consistent with its role in enhancing fat oxidation and lipid metabolism, outcomes that are particularly important given the established links between visceral adiposity, systemic

inflammation, and insulin resistance in T2DM. By contrast, participants in the HIIT group demonstrated measurable increases in lean mass, suggesting potential anabolic adaptations through the recruitment of type II muscle fibres and stimulation of protein synthesis pathways during repeated high-intensity bouts. Such adaptations may be clinically relevant for older adults or those with sarcopenic obesity, where the preservation of muscle mass is vital for functional independence and metabolic resilience.

These findings should, however, be interpreted with caution. Increases in lean mass of this magnitude are uncommon in this population over short intervention periods, and the use of bioelectrical impedance analysis (BIA) may have contributed measurement variability, as results can be influenced by hydration status and other physiological factors. While our observations are directionally consistent with previous reports showing modest hypertrophic effects of high-intensity exercise [45,46], the potential influence of methodological limitations cannot be excluded. Taken together, these results suggest complementary benefits of both modalities: A+R is more effective for reducing adiposity, while HIIT may contribute to lean mass improvements. Future studies employing more robust assessment methods such as DXA or MRI are warranted to validate these outcomes.

## Implications for exercise prescription

The current findings reinforce the importance of adopting a tailored approach to exercise prescription for individuals with T2DM. Both HIIT and A+R demonstrated clinically significant benefits, but their respective advantages suggest that different patient profiles may be better suited to one modality over the other. Importantly, this study affirms that structured physical activity when appropriately matched to individual needs and capacities can offer multifaceted improvements in metabolic, physical, and psychosocial outcomes.

**HIIT** is ideally suited for patients who require or prefer time-efficient interventions to address glycaemic dysregulation and insulin resistance. Its efficacy in rapidly improving FG and HOMA-IR, alongside its capacity to promote muscle hypertrophy, positions HIIT as a powerful tool in metabolic rehabilitation particularly among younger adults, those with limited time availability, or patients prioritising improvements in insulin sensitivity and lean body mass. However, HIIT's demanding nature may pose adherence challenges for certain populations, including older adults or those with comorbidities.

**Combined Aerobic and Resistance Training (A+R)** is recommended for individuals seeking more holistic, gradual improvements in cardiovascular health, glycaemic control, body composition, and quality of life. The moderate intensity of this intervention enhances long-term adherence and appears to offer broader psychosocial benefits, particularly in the domains of psychological and social well-being. A+R may be more suitable for patients at earlier stages of diabetes progression or those with physical or psychological barriers to engaging in high-intensity protocols.

Clinicians should consider not only physiological outcomes but also patient preferences, accessibility, perceived enjoyment, and risk profiles when recommending exercise regimens. These findings lend further support to the development of personalised, patient-centred strategies in diabetes rehabilitation.

## Study strengths and limitations

This study offers robust evidence for the comparative efficacy of two distinct exercise modalities within a randomised controlled design. The integration of both subjective and objective outcomes enhances the ecological validity of the findings, while the use of a rigorous statistical analysis adds methodological rigour.

However, certain limitations must be acknowledged: The findings are based on participants who adhered to their prescribed regimens, which may limit generalisability to patients with advanced diabetic complications. The study duration was limited to 12 weeks; longer-term follow-up is necessary to assess sustainability and the durability of observed effects on body composition and glycaemic control. Real-world implementation may vary, particularly with HIIT, which requires higher effort and supervision that may not always be feasible in unsupervised or community settings.

### Directions for future research

Future investigations should aim to:

**Explore combination regimens**: Assessing whether integrating HIIT with A+R in phased or concurrent formats leads to additive or synergistic effects.

**Evaluate long-term outcomes**: Trials extending beyond six months will be critical to understanding chronic adaptations in glycaemic control, cardiovascular fitness, and patient adherence.

**Conduct subgroup analyses**: Examining the differential responses in various T2DM populations, such as older adults, individuals with multiple comorbidities, or those at high cardiovascular risk, will offer deeper insights into personalised exercise planning.

These research directions will further inform clinical guidelines and enhance the translation of evidence-based exercise strategies into sustainable diabetes care models.

## Conclusion

This 12-week randomised controlled trial demonstrated that both combined aerobic–resistance training (A+R) and high-intensity interval training (HIIT) significantly improved metabolic, functional, and quality-of-life outcomes in adults with type 2 diabetes mellitus compared with standard care. The two modalities conferred distinct advantages: HIIT was superior for fasting glucose reduction and muscle mass gains, whereas A+R produced greater improvements in HbA1c, adiposity reduction, and multiple domains of quality of life. Although no consistent statistically significant differences emerged between A+R and HIIT across all outcomes, the pattern of findings supports the use of personalised exercise prescriptions tailored to specific therapeutic goals, whether metabolic control, functional improvement, or psychosocial benefit. The results should be interpreted considering study limitations, including reliance on compliant participants, short intervention duration, and absence of dietary control. These findings nonetheless reinforce the role of structured exercise as a cornerstone of T2DM management and suggest that hybrid or flexible training models may provide a pragmatic solution to address diverse patient needs and preferences. Future trials of longer duration, in more heterogeneous populations, and with controlled diet or combined behavioural strategies, are warranted to establish the sustainability and generalisability of these benefits.

## Supporting information

**S1 File. Intervention Characteristics by Group (FITT-VP Framework) [35].**
(DOCX)

**S2 File. CONSORT –Checklist.**
(PDF)

**S3 File. Protocol.**
(PDF)

## Acknowledgments

The authors express their heartfelt gratitude to all participants who willingly taken part in this study. Their invaluable contributions and dedication have made this research possible. Their cooperation and commitment were deeply appreciated.

## Author contributions

**Conceptualization:** Sampath Kumar Amaravadi, Patrícia dos Santos Vigário.

**Data curation:** Sampath Kumar Amaravadi.

**Formal analysis:** Sampath Kumar Amaravadi, Arthur de Sá Ferreira, Patrícia dos Santos Vigário.

**Investigation:** Sampath Kumar Amaravadi, Patrícia dos Santos Vigário.

**Methodology:** Sampath Kumar Amaravadi, Arthur de Sá Ferreira, Patrícia dos Santos Vigário.

**Project administration:** Patrícia dos Santos Vigário.

**Resources:** Sampath Kumar Amaravadi.

**Software:** Arthur de Sá Ferreira.

**Supervision:** Arthur de Sá Ferreira, Patrícia dos Santos Vigário.

**Validation:** Sampath Kumar Amaravadi, Arthur de Sá Ferreira, Patrícia dos Santos Vigário.

**Visualization:** Sampath Kumar Amaravadi, Arthur de Sá Ferreira.

**Writing – original draft:** Sampath Kumar Amaravadi.

**Writing – review & editing:** Arthur de Sá Ferreira, Patrícia dos Santos Vigário.

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
