## [Decision Letter · Decision Letter 0]

17 Sep 2025

Dear Dr. Amaravadi,

Thank you for submitting your manuscript to PLOS ONE. After careful consideration, we feel that it has merit but does not fully meet PLOS ONE’s publication criteria as it currently stands. Therefore, we invite you to submit a revised version of the manuscript that addresses the points raised during the review process.

Dear authors, as you can see, the reviewers have requested substantial revisions to your manuscript. We are certainly willing to reconsider a revised submission, but please know that this is not preliminary acceptance of your paper. When returning your revised manuscript, please be sure to include a point-by-point summary of the suggestions of the reviewers that specifies how and where in the text you have addressed the suggestions.

We look forward to receiving your revised manuscript.

Kind regards,

Ricardo Ney Oliveira Cobucci, Ph.D

Academic Editor

PLOS ONE

Journal Requirements:

2. We note that you have selected “Clinical Trial” as your article type. PLOS ONE requires that all clinical trials are registered in an appropriate registry (the WHO list of approved registries is at https://www.who.int/clinical-trials-registry-platform/network/primary-registries"
https://www.who.int/clinical-trials-registry-platform/network/primary-registries and more information on trial registration is at http://www.icmje.org/about-icmje/faqs/clinical-trials-registration/). Please state the name of the registry and the registration number (e.g. ISRCTN or ClinicalTrials.gov) in the submission data and on the title page of your manuscript. a) Please provide the complete date range for participant recruitment and follow-up in the methods section of your manuscript. b) If you have not yet registered your trial in an appropriate registry, we now require you to do so and will need confirmation of the trial registry number before we can pass your paper to the next stage of review. Please include in the Methods section of your paper your reasons for not registering this study before enrolment of participants started. Please confirm that all related trials are registered by stating: “The authors confirm that all ongoing and related trials for this drug/intervention are registered”. Please see http://journals.plos.org/plosone/s/submission-guidelines#loc-clinical-trials for our policies on clinical trials.

3. We note that your Data Availability Statement is currently as follows: All relevant data are within the manuscript and its Supporting Information files.]

Reviewers' comments:

Reviewer's Responses to Questions

**Comments to the Author**

1. Is the manuscript technically sound, and do the data support the conclusions?

Reviewer #1: Yes

Reviewer #2: No

2. Has the statistical analysis been performed appropriately and rigorously?

Reviewer #1: Yes

Reviewer #2: Yes

3. Have the authors made all data underlying the findings in their manuscript fully available?

Reviewer #1: Yes

Reviewer #2: Yes

4. Is the manuscript presented in an intelligible fashion and written in standard English?

Reviewer #1: Yes

Reviewer #2: Yes

Reviewer #1: I have some comments attached to the form. Please find and revise based on it:

1. Is the Title original, and does it provide a complete and accurate description of the article's content?

To strengthen it, consider adding specificity on the duration (e.g., "12-Week") for better searchability.

2. Does the Abstract provide a complete and accurate description of the article's content?

For improvement, include effect sizes or confidence intervals in the results section to align with modern reporting standards seen in recent meta-analyses, providing readers with a quicker grasp of clinical significance.

3. Are study problems and gaps clearly stated based on the recent studies in the introduction section?

To enhance, explicitly discuss gaps in long-term or diverse population studies:

- Badri Al-mhanna S, Wan Ghazali W S, Batrakoulis A, Alkhamees N H, Drenowatz C, Mohamed M, et al . The Impact of Various Types of Exercise on Lipid Metabolism in Patients with Type 2 Diabetes and Concurrent Overweight/Obesity: A Narrative Review. Ann Appl Sport Sci 2024; 12. 10.61186/aassjournal.1324

- Tayebi, S.M., Bagherian, P., Bassami, M., Basereh, A. and Ahmadabadi, S. (2025), Impact of a 12-week High-Intensity Interval Training With Spirulina Supplementation on Insulin Resistance-Mediated by Apo-A, -B, and -J in Men With Obesity. Eur J Sport Sci, 25: e12285. https://doi.org/10.1002/ejsc.12285

4. Is the rationale/justification for conducting the study clearly stated in the introduction section?

Strengthen by adding a sentence on potential cost-effectiveness or accessibility:

- Tayebi S M, Saeidi A, Shahghasi R, Golmohammadi M. The Eight-Week Circuit Resistance Training Decreased the Serum Levels of WISP-1 and WISP-2 in Individuals with Type 2 Diabetes. Ann Appl Sport Sci 2023; 11 (4). 10.61186/aassjournal.1290

- Tayebi, S. M., Motaghinasab, S., Eslami, R., Ahmadabadi, S., Basereh, A., & Jamhiri, I. (2024). Impact of 8-week cold-and warm water swimming training combined with cinnamon consumption on serum METRNL, HDAC5, and insulin resistance levels in diabetic male rats. Heliyon, 10(8). 10.1016/j.heliyon.2024.e29742

5. Are the study aims and novelty clearly stated in the introduction section, and are they logical?

To bolster novelty, contrast with prior studies focusing only on metabolic markers, emphasizing the inclusion of QoL:

- Cobo-Mejía E A, Castellanos-Vega R D P. The Effects of Physical Activity on Glycated Hemoglobin and Quality of Life in Adults with Diabetes Mellitus: A Systematic Review of the Literature and Meta-Analysis. Ann Appl Sport Sci 2024; 12 (4). 10.61186/aassjournal.1420

- Tayebi S M, Eslami R, Iranshad I, Golmohammadi M. The Effect of Eight Weeks of Circuit Resistance Training on Serum Levels of GPR119 and β-Arrestin1 in Individuals with Type 2 Diabetes. Ann Appl Sport Sci 2023; 11 (3). 10.61186/aassjournal.1283

6. In the materials and methods section, is the study design robust and appropriate to the stated aim?

Suggest including blinding details (e.g., assessor blinding) to mitigate bias.

7. Are the methods described (like Participants/Patients/Animals, Exercise Training Protocol/ Study Tools and Assessments, Data Collection Protocol, Statistical Analysis, etc.) in sufficient detail and standard methods to reproduce the experiment?

For enhancement, provide more specifics on HIIT intensity monitoring (e.g., heart rate devices) and progression criteria.

8. In the results section of the full text, are the contents reported in a manner that is consistent with reporting guidelines for the article/study type?

To improve, add effect sizes (Cohen's d) for all outcomes, as done in the primary analysis description, to better quantify clinical relevance.

9. Is the discussion section critical and comprehensive with a robust, valid, and reliable data interpretation (analyze the proper findings presentation, Logical Comparison with the Related Literature, Interpretation of the Findings, and the Mechanisms of these Findings)?

Strengthen by discussing potential confounders (e.g., diet) and integrating more 2024-2025 Stidies. I sugessted some updated and related references above. Use and cite them.

10. Are the conclusions drawn supported by the data, or need to be adjusted with new revisions?

No major revisions needed, but temper claims on "broader benefits" by noting non-significant between-group differences in some outcomes.

11. Are the applicable remarks prescribed based on the exact findings of this study, or do they need to be adjusted with new revisions?

Remarks on personalized prescriptions are directly based on findings (e.g., modality-specific strengths). Implications for clinicians are practical and evidence-based. Adjust slightly to emphasize adherence limitations, as results apply to compliant patients, and suggest hybrid models for future.

12. Are statements supported appropriately by citations?

Add citations for all mechanistic claims to enhance rigor. I sugessted some updated and related references above. Use and cite them.

13. Are the references appropriate in number and up-to-date?

36 references are appropriate for an RCT; many are recent (2020-2025, e.g., IDF 2025, Gajanand 2024), aligning with knowledge updates. Include more 2024-2025 meta-analyses for HIIT vs. combined training to ensure currency. I sugessted some updated and related references above. Use and cite them.

14. Are figures and tables well-constructed and of sufficiently high resolution (i.e., not blurry)?

All seems ok.

15. Are figures and tables well-annotated and easy to read and interpret?

All seems ok.

16. Is the writing clear, concise, and logical?

Writing is generally clear and logical, with structured flow. Some sentences are wordy (e.g., introduction repetitions on exercise benefits). Concise overall, but edit for grammar (e.g., "preventing of type-2" to "prevention of type 2"). This would make it more engaging and professional.

17. Does the writing impede scientific meaning or cause confusion?

No major impediments, but minor inconsistencies (e.g., "A+R" vs. "combined aerobic and resistance") and awkward phrasing (e.g., "normalizes in the prevention") could confuse. Revise for precision to avoid ambiguity, ensuring scientific accuracy.

Reviewer #2: First, thanks to the editor for the opportunity to revise the manuscript entitled “Comparative Effects of Combined Aerobic-Resistance Training, and High-Intensity Interval Training on Insulin Resistance, Glycaemic Control, Body Composition, and Quality of Life in Patients with Type 2 Diabetes Mellitus: A Randomised Controlled Trial”.

This study addresses the effects of combined aerobic-resistance training and high-intensity interval training, compared to a control group, on insulin resistance, glycaemic control, body composition, physical function, and quality of life in adults with T2DM. While the topic is timely and relevant, several aspects need clarification to better highlight the study’s novelty, enhance methodological transparency, and ensure the research question is thoroughly addressed.

Please find below my detailed comments and suggestions.

ABSTRACT

- If a control group is included, it should be explicitly mentioned in the study aim in the abstract. Please, the authors should clarify whether they are testing the superiority of exercise interventions over control, and/or differential effects between the two exercise modalities.

- The primary outcomes displayed in the abstract are different from those in the methods section or study protocol (fasting blood sugar vs. fasting insulin). This sounds misleading, as it seems that the research question is not clear. I suggest the authors should clarify this point.

- I suggest removing the sentence “Significant group × time interactions (p < 0.001) were found across most outcomes”. This statement lacks specificity and does not aid interpretation. Consider removing or replacing it with concrete effect estimates and comparisons.

- The abstract would benefit from greater focus on between-group comparisons giving mean differences with confidence intervals and P values, as this aligns with the core research question of whether the type of exercise (A+R vs. HIIT) differentially insulin resistance, glycaemic control, body 24 composition, physical function, and quality of life. Highlighting whether A+R produced superior or distinct outcomes relative to HIIT would provide readers with a clearer understanding of the study's contribution.

- The conclusions should explicitly state whether one intervention was superior, equivalent, or non-inferior to the others, based on between-group analyses.

INTRODUCTION

- Overall, I suggest rewriting and dividing the information introduced within the introduction in more paragraphs. As it is now presented, it is not clear and is difficult to read. There are some terms which do not apply to the outcomes specified in this study, and this may distract the reader. As a suggestion, authors could restructure the information as follows: (1) public health significance of T2DM and cardiometabolic disease; (2) overview of exercise modalities (HIIT vs A+R) and knowledge gaps; (3) limitations in prior research (e.g., lack of direct comparison trials, reliance on surrogate outcomes); and (4) the study's aim and hypothesis with supporting rationale.

- The authors have abbreviated the words triglyceride (TGL) and high-density lipoprotein (HDL), but then they do not appear again within the text. Moreover, there are some words which have been abbreviated twice as “HIIT” or “T2DM”, this is a problem when do not reviewing the text after using Artificial Intelligence. Please review carefully the whole manuscript and address these concerns.

- The hypotheses are stated, but they would be strengthened by pre-introducing relevant mechanistic or empirical evidence. For example, why HIIT is expected to enhance insulin sensitivity and muscle mass, and why A+R might better improve glycemic control, fat loss, or quality of life.

- Line 119: Consider defining HbA1c and HOMA-IR upon first mention, as readers who are not specialists in endocrinology or metabolism may not be familiar with these terms. Given that this journal targets a broad scientific audience, it is important to maintain clarity and accessibility for all readers.

METHODS

- Please indicate whether the study was conducted in accordance with the CONSORT guidelines for reporting randomized controlled trials. Note that an updated version was recently published (https://doi.org/10.1038/s41591-025-03635-5).

- The term “UAE” is not defined in the manuscript. Please clarify what this acronym refers to, especially in the context of the study setting.

- Within methods section, the authors introduce Group A, B, and C without prior explanation or alignment with the terminology used to describe the interventions (e.g., A+R, HIIT, control). For consistency and clarity, I recommend using the same terms throughout the manuscript.

- The Intervention section is challenging to follow due to its unsystematic organization. For clarity and precision, I recommend restructuring it using the FITT‑VP framework, detailing, for each group, the Frequency, Intensity, Time, Type, Volume, and Progression of the exercises. This approach will significantly improve readability and enhance reproducibility.

- The adverse events subsection contains outcome data that should be presented in the Results section. I recommend relocating this content to ensure proper structural alignment with CONSORT guidelines.

- Did the authors ensure consistency in the timing of exercise sessions across participants and groups? When exactly were these sessions performed? This is particularly relevant in individuals with T2DM, as accumulating evidence suggests that afternoon or evening exercise may enhance glycemic control more effectively than morning exercise (PMID: 37226675; 30426166; 37732475). Addressing this could help interpret the observed glycemic outcomes more accurately.

- Line 197: Please rephrase to indicate that the 6MWT is the assessment tool, and the outcome is the distance walked in meters or functional aerobic capacity.

- Line 198: Please, provide device and brand to assess body composition.

- Line 198 and whole manuscript: I suggest rephrasing “fasting blood sugar” to “fasting glucose”, to be more specific with the terminology in the medical field.

- Were conditions of assessment standardized? For example, fasting conditions, amount of physical activity the day in which blood draws were conducted and the days before, if they were in the morning or not, etc. Please provide more information.

- How were waist and hip circumference assessed? Did these measurements follow the International Society for the Advancement of Kinanthropometry (ISAK) procedures? Please provide more information.

- Which statistical test was used to compare the groups at baseline? Please specify.

RESULTS

- I suggest authors should start the results section describing well the flow-chart and whether some people were lost to follow-up at post-assessment. Moreover, they should specify whether they followed an intention-to-treat analysis or per-protocol.

- Please report in Table 1 the number and percentage of participants who were taking oral hypoglycaemic agents and/or insulin therapy.

- Please provide in Table 1 the baseline values for each study group across the main outcome domains (e.g., glycemic control, body composition, quality of life). This additional information would improve transparency and help readers better understand the characteristics and comparability of the study sample at baseline.

- The results emphasize within-group changes but lack formal between-group comparisons, which are essential for interpreting the relative efficacy of A+R versus HIIT and control. The authors should report between-group mean differences with confidence intervals and P-values to properly support any claims of superiority. Without these, conclusions about which intervention is more effective may be misleading. Moreover, this would simplify this section and improve readability.

- Line 302: What does “large effect sizes” mean? The results section should be specific.

- In Table 3, please specify the units of measurement for all outcomes. For example, it is unclear what metric GPAQ refers to, minutes of activity per week, MET-min/week, or another index, making interpretation difficult.

- As fasting insulin is listed as the primary outcome in the Methods section, it would be more appropriate to present it first in Table 3 to reflect its importance in the study design and maintain consistency.

- In the introduction and throughout the manuscript, the authors suggest that greater improvements in glycemic control and body composition were expected in the exercise groups compared to the control group. However, Table 3 does not clearly convey these between-group differences. For instance, it is difficult to determine whether fasting insulin significantly decreased more in the HIIT group relative to control. This could be improved by presenting the between-group differences in a consistent and intuitive direction (e.g., exercise group minus control group), and clearly marking which comparisons are statistically significant.

- Please provide data about adherence to the exercise interventions (e.g., attendance to exercise sessions) and compliance (e.g., percentage of sessions in which the participants met the intensity prescribed). This would help to improve interpretability of the results.

Were there any missing data at baseline or at follow-up assessments? If so, the extent and pattern of missing values should be clearly reported. In Table 1, any missing values at baseline should be explicitly displayed (e.g., using “n (%)” format or footnotes), and similar reporting should be applied in subsequent tables.

DISCUSSION

- Lines 314-317: The statement that both exercise groups “significantly improved” various outcomes lack clarity. Improved relative to what—baseline values, the control group, or each other? Please specify the comparison group and clarify whether these improvements are based on within-group changes or between-group comparisons. It is essential to frame these findings clearly to answer the central research question of whether one intervention is superior.

- Lines 322–330: I recommend a more cautious interpretation of the findings. As no statistically significant differences were found between A+R and HIIT groups in the key outcomes, this paragraph currently overstates the conclusions. The authors should revise the narrative to focus on the lack of superiority between interventions, ensuring that the interpretation aligns with the results from between-group comparisons.

- Lines 407-426: The description of the HIIT intervention (whether it involved primarily aerobic or resistance-based components) remains unclear in the Methods section and should be clarified. This point is particularly relevant given the reported 8 kg increase in muscle mass over just 12 weeks, which appears unusually large for this population and timeframe. The authors should justify these results more carefully and consider whether the body composition measurement technique may have influenced the outcome (e.g., potential limitations or biases in the assessment tool). Providing additional methodological detail and contextualizing the findings in relation to prior literature would help improve credibility and interpretation.

- I suggest reconsidering the use of a separate subsection for “HbA1c Control.” The content could be more effectively integrated into the main discussion of glycemic outcomes to improve flow and readability, especially given the interconnected nature of the results.

- In the “Body Composition Adaptations” subsection, there appears to be some repetition of information. I recommend summarizing overlapping content and considering a more integrated structure to enhance clarity and reduce redundancy.

- I suggest removing the “Physical Exercise and Inflammation” subsection, as it appears unrelated to the primary outcomes and overall scope of the manuscript.

**Do you want your identity to be public for this peer review?** For information about this choice, including consent withdrawal, please see our Privacy Policy

Reviewer #1: No

Reviewer #2: **Yes: ** Antonio Clavero-Jimeno

---

## [Author Response · Author response to Decision Letter 1]

23 Sep 2025

We would like to thank the reviewers and editor for their thoughtful and constructive feedback. We have carefully revised the manuscript in response to all comments, which we believe has enhanced its clarity, rigour, and overall impact. Please refer to the uploaded PDF in the “Response to Reviewers” section, where each comment is followed by our detailed response and the corresponding page and line numbers indicating where revisions were made in the revised manuscript.

---

## [Decision Letter · Decision Letter 1]

29 Sep 2025

Dear Dr. Amaravadi,

Thank you for submitting your manuscript to PLOS ONE. After careful consideration, we feel that it has merit but does not fully meet PLOS ONE’s publication criteria as it currently stands. Therefore, we invite you to submit a revised version of the manuscript that addresses the points raised during the review process.

Provide new corrections with track changes.

We look forward to receiving your revised manuscript.

Kind regards,

Ricardo Ney Oliveira Cobucci, Ph.D

Academic Editor

PLOS ONE

Journal Requirements:

Reviewers' comments:

Reviewer's Responses to Questions

**Comments to the Author**

Reviewer #1: All comments have been addressed

Reviewer #2: All comments have been addressed

2. Is the manuscript technically sound, and do the data support the conclusions?

Reviewer #1: Yes

Reviewer #2: Yes

3. Has the statistical analysis been performed appropriately and rigorously?

Reviewer #1: Yes

Reviewer #2: Yes

4. Have the authors made all data underlying the findings in their manuscript fully available?

Reviewer #1: Yes

Reviewer #2: Yes

5. Is the manuscript presented in an intelligible fashion and written in standard English?

Reviewer #1: Yes

Reviewer #2: Yes

Reviewer #1: No more revision is needed.

All comments covered in revised version.

It is good for publication now.

Reviewer #2: The authors have done a good job addressing the comments from the first round of review. However, a few issues remain that would benefit from clarification or further refinement in the current version of the manuscript:

- Results section: Please provide mean differences between groups for all outcomes, as this aligns with the study’s primary research question of superiority testing. Reporting within-group changes is unnecessary and may be misleading; it is sufficient to include these in the tables. For example: Fasting glucose: The HIIT group (mean difference: –X; 95% CI, X to X) and the A+R group (mean difference: –X; 95% CI, X to X) exhibited significantly greater reductions compared with the control group (Table 3).

- Abstract: The same principle should be applied in the abstract, ensuring consistency with the research question.

- My previous comments on baseline characteristics may have been misunderstood. My main suggestion was to report the total sample size available for each outcome at baseline, as this helps identify any missing values and ensures compliance with CONSORT 2025 guidelines. Based on the flowchart, it appears that no missing values are present, so this is not needed. In addition, I indicated that if baseline comparisons are included, the statistical test used should be specified. However, as the authors also correctly note, statistical testing of baseline differences is not recommended in RCTs; therefore, I suggest removing this information from the tables and manuscript to avoid confusion.

**Do you want your identity to be public for this peer review?** For information about this choice, including consent withdrawal, please see our Privacy Policy

Reviewer #1: No

Reviewer #2: **Yes: ** Antonio Clavero-Jimeno

---

## [Author Response · Author response to Decision Letter 2]

30 Sep 2025

RESPONSE TO REVIEWER

We thank the reviewers for their detailed feedback and for recognizing the improvements made in the revised manuscript. We are pleased that Reviewer #1 finds the work suitable for publication.

We have carefully addressed the additional clarifications requested by Reviewer #2. Below, we provide a point-by-point response, with tracked changes references to the revised manuscript.

Reviewer #2

Comment 1: Results section

“Please provide mean differences between groups for all outcomes, as this aligns with the study’s primary research question of superiority testing. Reporting within-group changes is unnecessary and may be misleading; it is sufficient to include these in the tables.”

Response:

We have revised the Results to focus exclusively on between-group mean differences (MDs) with 95% CIs, consistent with superiority testing. All within-group changes have been removed from the narrative and retained only on tables.

Change made: For example, the Results now read (Page 12, Lines 310-353):

“The HIIT group demonstrated a greater reduction in fasting glucose compared with control (MD −29.1 mg/dL; 95% CI −41.2 to −17.0), and the A+R group also improved relative to control (MD −20.6 mg/dL; 95% CI −31.0 to −10.2) (Table 3).”

Other outcomes: HbA1c, fasting insulin, HOMA-IR, body composition, functional capacity, and quality of life have all been updated in the same way.

Comment 2: Abstract

“The same principle should be applied in the abstract, ensuring consistency with the research question.”

Response:

We have revised the Abstract to report intervention vs control MDs with 95% CIs for the primary and secondary outcomes, ensuring consistency with the Results.

Change made: Abstract, Results section (Page 2-3, Lines 36–49): “Compared with control, the HIIT group reduced fasting glucose (MD −29.1 mg/dL; 95% CI −41.2 to −17.0) and A+R also improved (MD −20.6 mg/dL; 95% CI −31.0 to −10.2). HbA1c was lower in HIIT (MD −3.35%; 95% CI −4.11 to −2.58) and A+R (MD −3.33%; 95% CI −4.03 to −2.62) compared with control…”

Comment 3: Baseline characteristics

“My main suggestion was to report the total sample size available for each outcome at baseline … Based on the flowchart, it appears that no missing values are present, so this is not needed. … I suggest removing this information from the tables and manuscript to avoid confusion.”

Response:

We confirm that all participants completed baseline assessments with no missing data, as shown in the flow diagram. Following the reviewer’s advice:

Change made in Tables 1–2: All baseline statistical tests (ANOVA/χ²) and p-values have been removed. Only descriptive values (mean ± SD or n [%]) are presented. (Page 11-12, Table 1&2).

Change made in Methods (Statistics): Clarified that baseline characteristics are presented descriptively, with no hypothesis testing (Page 10, Lines 259–261).

We believe these revisions fully address Reviewer #2’s comments. All requested clarifications have been incorporated, with tracked-changes visible in the revised manuscript (clean and marked versions uploaded). We hope the manuscript is now acceptable for publication.

With thanks,

Sincerely,

Sampath

---

## [Decision Letter · Decision Letter 2]

9 Oct 2025

Dear Dr. Amaravadi,

Thank you for submitting your manuscript to PLOS ONE. After careful consideration, we feel that it has merit but does not fully meet PLOS ONE’s publication criteria as it currently stands. Therefore, we invite you to submit a revised version of the manuscript that addresses the points raised during the review process.

We look forward to receiving your revised manuscript.

Kind regards,

Ricardo Ney Oliveira Cobucci, Ph.D

Academic Editor

PLOS ONE

Journal Requirements:

Additional Editor Comments (if provided):

Reviewers' comments:

Reviewer's Responses to Questions

**Comments to the Author**

Reviewer #2: All comments have been addressed

Reviewer #3: (No Response)

2. Is the manuscript technically sound, and do the data support the conclusions?

Reviewer #2: Yes

Reviewer #3: Partly

3. Has the statistical analysis been performed appropriately and rigorously?

Reviewer #2: Yes

Reviewer #3: No

4. Have the authors made all data underlying the findings in their manuscript fully available?

Reviewer #2: Yes

Reviewer #3: Yes

5. Is the manuscript presented in an intelligible fashion and written in standard English?

Reviewer #2: Yes

Reviewer #3: Yes

Reviewer #2: No further comments are needed. The authors have addressed all concerns, and the manuscript has been substantially improved.

Reviewer #3: The assumptions used for the sample size calculation need to be clinically justified.

P-values should be adjusted to account for the large number of primary and secondary outcomes analyzed compared to the small sample size.

In Table 3, add within-group changes. Remove p-values for “interaction” and “between-group” effects, as they are not meaningful for direct between-group comparisons. Cohen’s d can be omitted, as mean differences are more clinically interpretable.

Many baseline variables appear to be imbalanced, including Age, T2DM duration, FG, PPBS, WHOQOL measures, sedentary time, overweight, eye complications and etc.. These imbalances raise concerns about the adequacy of randomization. Further, the linear mixed model does not specify covariate adjustment. These imbalanced baseline variables should be addressed.

**Do you want your identity to be public for this peer review?** For information about this choice, including consent withdrawal, please see our Privacy Policy

Reviewer #2: **Yes: ** Antonio Clavero-Jimeno

Reviewer #3: No

---

## [Author Response · Author response to Decision Letter 3]

12 Oct 2025

Response to reviewer 3 comments

Comment 1:

Many baseline variables appear to be imbalanced, including age, T2DM duration, FG, PPBS, WHOQOL measures, sedentary time, overweight, eye complications, etc. These imbalances raise concerns about the adequacy of randomisation. Further, the linear mixed model does not specify covariate adjustment. These imbalanced baseline variables should be addressed.

Authors Response:

We thank the reviewer for this valuable observation. We acknowledge that several baseline variables demonstrated numerical differences between groups despite randomisation. Given the modest sample size (n = 30 per group), these imbalances likely occurred by chance. To address this concern and ensure that between-group effects reflect intervention-related changes rather than pre-existing differences, the linear mixed-effects models were re-specified to include key baseline covariates.

The revised model included age, duration of type 2 diabetes mellitus (T2DM), baseline fasting glucose (FG), and baseline WHOQOL-physical domain scores as covariates. This covariate-adjusted model was applied to all primary and secondary outcomes. The Statistical Analysis section has been rewritten accordingly, and the results presented in Table 3 are now explicitly described as covariate-adjusted mean differences (95 % CI).

A statement has also been added to Table 1 legend clarifying that observed baseline imbalances were subsequently adjusted for in the mixed-model analysis. – Page number 12.

Comment 2:

In Table 3, add within-group changes. Remove p-values for “interaction” and “between-group” effects, as they are not meaningful for direct between-group comparisons. Cohen’s d can be omitted, as mean differences are more clinically interpretable.

Authors Response:

We fully agree with the reviewer’s suggestion. Table 3 has been revised to include within-group changes (Post – Pre) for each intervention arm (HIIT, A+R, and Control). All p-values for interaction and between-group effects, as well as Cohen’s d values, have been removed. The table now reports covariate-adjusted between-group mean differences (95 % CI) for greater clinical interpretability. Correspondingly, references to effect size calculations have been removed from the Statistical Analysis section. - Page numbers 15 & 16

Comment 3:

The assumed effect size (f = 0.30) used in the sample size calculation requires justification.

Authors Response:

We have now added a clear clinical and empirical justification for the assumed moderate effect size (f = 0.30) within the Sample Size Calculation subsection. This assumption was based on prior exercise intervention trials in type 2 diabetes demonstrating moderate changes in HOMA-IR and HbA1c (Bird & Hawley, 2016; Jelleyman et al., 2015; Amaravadi et al., 2024). The text has been revised as follows:

“The assumed moderate effect size (f = 0.30) was based on previous randomised controlled trials in patients with type 2 diabetes reporting moderate improvements in HOMA-IR and HbA1c following structured exercise interventions (Bird & Hawley, 2016; Jelleyman et al., 2015; Amaravadi et al., 2024). This was considered a conservative and clinically meaningful estimate.” - Page number -10

We thank the reviewer for their thoughtful feedback, which has substantially improved the methodological transparency, statistical rigour, and clinical interpretability of our manuscript.

On behalf of all authors,

Sampath Amaravadi

---

## [Decision Letter · Decision Letter 3]

21 Oct 2025

Dear Dr. Amaravadi,

Thank you for submitting your manuscript to PLOS ONE. After careful consideration, we feel that it has merit but does not fully meet PLOS ONE’s publication criteria as it currently stands. Therefore, we invite you to submit a revised version of the manuscript that addresses the points raised during the review process.

We look forward to receiving your revised manuscript.

Kind regards,

Ricardo Ney Oliveira Cobucci, Ph.D

Academic Editor

PLOS ONE

Journal Requirements:

Reviewers' comments:

Reviewer's Responses to Questions

**Comments to the Author**

Reviewer #3: (No Response)

2. Is the manuscript technically sound, and do the data support the conclusions?

Reviewer #3: (No Response)

3. Has the statistical analysis been performed appropriately and rigorously?

Reviewer #3: (No Response)

4. Have the authors made all data underlying the findings in their manuscript fully available?

Reviewer #3: (No Response)

5. Is the manuscript presented in an intelligible fashion and written in standard English?

Reviewer #3: (No Response)

Reviewer #3: P-values should be adjusted to account for the large number of primary and secondary outcomes analyzed compared to the small sample size.

**Do you want your identity to be public for this peer review?** For information about this choice, including consent withdrawal, please see our Privacy Policy

Reviewer #3: No

---

## [Author Response · Author response to Decision Letter 4]

24 Oct 2025

Response to Reviewer #3

Reviewer #3 Comment:

P-values should be adjusted to account for the large number of primary and secondary outcomes analyzed compared to the small sample size.

Authors’ Response:

We sincerely thank the reviewer for this valuable observation and agree that adjustment for multiple comparisons is an important consideration in studies with several outcomes and a modest sample size.

We would like to clarify that baseline P-values were not presented in the first place because this practice is no longer recommended in randomized controlled trials. As previously discussed, this is a long-standing practice that has been widely discouraged in the literature (see:https://doi.org/10.1136/bmj.319.7203.185, http://dx.doi.org/10.31234/osf.io/qftwg, https://doi.org/10.2106/jbjs.21.01166). Any imbalance observed at baseline is likely due to randomization and does not warrant inferential testing (e.g., ANCOVA). Variables that are clinically or theoretically important were already included as covariates in the mixed-model analysis to account for any potential baseline variation.

However, we fully respect the reviewer’s concern regarding the potential for inflated Type I errors due to multiple comparisons. To address this, we have re-evaluated the analyses and adjusted all P-values for multiple testing using both the Benjamini–Hochberg False Discovery Rate (FDR, q = 0.05) and Bonferroni correction (m = 23 tests). The adjusted P-values are now reported in Table 1 and Table 3 to ensure full transparency and statistical robustness.

The Statistical Analysis section has been revised to include the following statement:

“Given the relatively small sample size and the number of primary and secondary outcomes, P-values were adjusted using both the Benjamini–Hochberg False Discovery Rate (FDR, q = 0.05) and Bonferroni correction (m = 23 tests; adjusted α = 0.002). Adjusted P-values are presented in the tables to account for multiple comparisons and ensure alignment with CONSORT 2025 recommendations.”

These revisions did not materially change the overall interpretation of the findings, as the principal outcomes and direction of effects remained consistent with the original analysis. We believe that these revisions further strengthen the methodological rigour, transparency, and reproducibility of the study, while respectfully addressing the reviewer’s concern.

Changes Made:

• Abstract – results section (Page 2-3) & Statistical Analysis section revised (Page 10).

• Adjusted P-values added to Table 1&3 (Pages 12-13).

• Results updated to reflect FDR and Bonferroni-adjusted P-values. (Page 11)

On behalf of all authors,

Sampath Amaravadi

---

## [Editor Report · Decision Letter 4]

3 Nov 2025

Comparative Effects of Combined Aerobic and Resistance Training Versus High-Intensity Interval Training on Insulin Resistance, Glycaemic Control, Body Composition and Quality of Life in Type 2 Diabetes: A 12-Week Randomised Controlled Trial

PONE-D-25-40562R4

Dear Dr. Amaravadi,

We’re pleased to inform you that your manuscript has been judged scientifically suitable for publication and will be formally accepted for publication once it meets all outstanding technical requirements.

Kind regards,

Ricardo Ney Oliveira Cobucci, Ph.D

Academic Editor

PLOS ONE
---

## [Editor Report · Acceptance letter]

PONE-D-25-40562R4

PLOS ONE

Dear Dr. Amaravadi,

I'm pleased to inform you that your manuscript has been deemed suitable for publication in PLOS ONE. Congratulations! Your manuscript is now being handed over to our production team.

Kind regards,

on behalf of

PROFESSOR Ricardo Ney Oliveira Cobucci

Academic Editor

PLOS ONE